



# Paramagnetic relaxivity of delocalized long-lived states of protons in chains of CH$_2$ groups

Aiky Razanahoera, Anna Sonnefeld, Geoffrey Bodenhausen, Kirill Sheberstov

Department of chemistry, École Normale Supérieure, PSL University, 75005 Paris, France

*Correspondence to*: Kirill Sheberstov, kirill.sheberstov@ens.psl.eu

**Abstract.** Long-lived states (LLS) have lifetimes $T_{LLS}$ that can be much longer than longitudinal relaxation times $T_1$. In molecules containing several geminal pairs of protons in neighbouring CH$_2$ groups, it has been shown that *delocalized* long-lived states can be excited by converting magnetization into imbalances between the populations of singlet and triplet states of each pair. Since the yield of conversion of observable magnetization into LLS and back are on the order of 10% or less if one

uses spin-locked induced crossing (SLIC), it would be desirable to boost the sensitivity by dissolution dynamic nuclear polarization (d-DNP). To enhance the magnetization of nuclear spins by d-DNP, the analytes must be mixed with radicals such as 4-hydroxy-2,2,6,6-tetramethylpiperidin-1-oxyl (TEMPOL) prior to freezing at low temperatures in the vicinity of 1 K. After dissolution, these radicals lead to an undesirable paramagnetic relaxation enhancement (PRE) which shortens not only the longitudinal relaxation times $T_1$ but also the lifetimes $T_{LLS}$ of long-lived states. It is confirmed in this work that PRE by

TEMPOL is less deleterious for LLS than for longitudinal magnetization for four different molecules: 2,2-dimethyl-2-silapentane-5-sulfonate (DSS), homotaurine, taurine, and acetylcholine. The relaxivities (i.e., the slopes of relaxation rates as a function of the radical concentration) of LLS $r_{LLS}$ are 3 to 5 times smaller than the relaxivities of longitudinal magnetization $r_1$. Partial delocalization of the LLS across neighbouring CH$_2$ groups may decrease this advantage, but in practice, this effect was observed to be minor when comparing taurine containing two CH$_2$ groups and homotaurine with three CH$_2$ groups.

Regardless of whether the LLS are delocalized or not, it is shown that PRE should not be a major problem for experiments combining d-DNP and LLS, provided the concentration of paramagnetic species after dissolution does not exceed 1 mM, a condition that is readily fulfilled in typical d-DNP experiments. In bullet d-DNP experiments however, it may be necessary to reduce TEMPOL by adding ascorbate or using lower concentrations of TEMPOL.

## Introduction

The lifetime of spin states in nuclear magnetic resonance (NMR) is normally limited by longitudinal relaxation. In certain cases, it is possible to access spin states that have extended lifetimes. Usually, these are associated with a coupled pair of spins with $I = \frac{1}{2}$ and correspond to population imbalances between singlet and triplet states of pairs of spins (Carravetta and Levitt, 2004; Carravetta et al., 2004). Such imbalances are also known as long-lived states (LLS). They are immune to intra-pair dipole-dipole interactions, which for pairs of protons are normally the dominant cause of longitudinal relaxation. The relaxation



time constants $T_{LLS}$ can be much longer than typical longitudinal relaxation time constants $T_1$. This feature is particularly useful for protein-ligand studies (Salvi et al., 2012; Buratto et al., 2014, 2016). Applications of LLS can be combined with different hyperpolarization methods, such as parahydrogen-based methods (Franzoni et al., 2012) or dissolution dynamic nuclear polarization (d-DNP) (Bornet et al., 2014; Kiryutin et al., 2019). D-DNP is the most universal method to achieve high spin polarization, and has found applications in drug screening (Lee et al., 2012; Kim et al., 2016), and in studies of metabolism by

*in-vivo* magnetic resonance imaging (MRI) (Nelson et al., 2013). Before dissolution, the saturation of the electron spin transitions by micro-wave irradiation of a solid sample near 1 K leads to an enhancement of the nuclear spin polarization by up to 4 orders of magnitude, compared to the thermal polarization at room temperature in the same magnetic field. The sample is then quickly dissolved and transferred to a solution-state NMR spectrometer, where the high-resolution signals are observed (Ardenkjær-Larsen et al., 2003). In an alternative approach known as "bullet DNP", the cold solid sample is ejected from the

polarizer and rapidly transferred to the NMR spectrometer where it is dissolved (Kouřil et al., 2019). After dissolution, the unpaired electrons of the dilute paramagnetic agent give rise to undesirable paramagnetic relaxation enhancement (PRE). For most molecules of interest, such as metabolites or potential drugs, proton relaxation is fast so that the level of hyperpolarization suffers during dissolution and transfer, which is the reason why d-DNP is more often used for $^{13}C$ or $^{15}N$ rather than for protons. Although molecules that are in enriched $^{13}C$ and $^{15}N$ offer many possibilities for the excitation of LLS (Feng et al., 2013; Elliott

et al., 2019; Sheberstov et al., 2019), there are several drawbacks of using heteronuclei. Labelled compounds are expensive and $^{13}C$ or $^{15}N$ NMR observation is much less sensitive compared to $^1H$. LLS involving pairs of protons often provide good contrast because protons are often directly exposed to the drug/target interface.

       Recently it was discovered that proton LLS involving geminal pairs of protons can be readily excited in many molecules containing at least two neighboring $CH_2$ groups (Sonnefeld et al., 2022a, b). Aliphatic chains, which are the focus

of this study, are commonly found in potential drugs, so that LLS of $CH_2$ groups could provide a breakthrough for drug screening using NMR. Hyphenation of LLS methodology with d-DNP offers promising perspectives, since at very low spin temperatures that are routinely achieved in d-DNP, singlet-triplet imbalances can result from a violation of the high-temperature approximation, so that LLS need not be excited by RF irradiation (Tayler et al., 2012; Bornet et al., 2014). LLS that involve chemically equivalent proton pairs in $CH_2$ groups need not be sustained by RF fields or protected by shuttling to

low fields. Therefore, one can transfer hyperpolarized samples to an NMR spectrometer for detection without significant losses of polarization. For small molecules, the ratios $T_{LLS}/T_1$ range typically from 2 to 6 for LLSs in $CH_2$ groups in non-degassed samples (Sonnefeld et al., 2022a). In this work, we carried out a systematic analysis of relaxivities, i.e., of the dependence of the relaxation rates of LLS and longitudinal magnetization on the concentration of the paramagnetic species 4-hydroxy-2,2,6,6-tetramethylpiperidin-1-oxyl (TEMPOL).






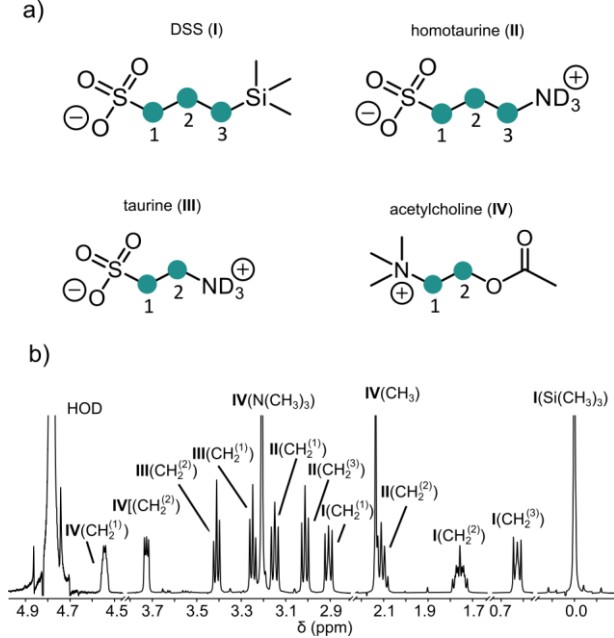

**Figure 1. (a) Chemical structures of four molecules supporting LLS of CH₂ groups studied in this work: 2,2-Dimethyl-2-silapentane-5-sulfonate sodium salt (DSS, I), homotaurine (II), taurine (III), acetylcholine (IV). CH₂ groups supporting LLS are numbered in each structure and highlighted by green circles. (b) Assignment of the ¹H NMR spectrum of a mixture containing all four compounds.**

Paramagnetic transition metal ions (Cu$^{2+}$, Mn$^{2+}$), lanthanides (Gd$^{3+}$) and triplet oxygen (O$_2$) have been shown to induce PRE of LLS, although PRE is not very efficient because the fluctuating external fields at the sites of two closely-spaced protons attached to the same carbon atom are strongly correlated (Tayler and Levitt, 2011). The effects of triplet oxygen on LLS have been further investigated (Erriah and Elliott, 2019). The question arises if fluctuating external fields due to the bulky TEMPOL radical are even more strongly correlated, in particular when they act on delocalized LLS involving several neighbouring CH$_2$ groups in molecules such as those shown in Figure 1. In DSS (I) and homotaurine (II), the LLS can be delocalized over all six protons of the three CH$_2$ groups, whereas in taurine (III) and acetylcholine (IV) the LLS always involves all four protons of both CH$_2$ groups. Titration experiments with TEMPOL allowed us to determine to what extent the radical affects the LLS lifetimes and to determine whether it is necessary to quench the radicals after dissolution (Miéville et al., 2010). In low fields, in particular after dissolution during the transfer between the polarizer and the NMR magnet, PRE may be exacerbated by translational diffusion (Borah and Bryant, 1981) of the paramagnetic molecules relative to the analytes (Miéville et al., 2011).

**Experimental methods**

The delocalised LLS were excited by using spin-lock induced crossing (SLIC) (DeVience et al., 2013) and its polychromatic extension (Sonnefeld et al., 2022b). A generic SLIC pulse sequence is illustrated in Figure 2a. After a non-selective 90º pulse that rotates the magnetization into the transverse plane, one, two or three continuous selective spin-lock





pulses with a common duration $\tau_{SLIC}$ are applied to the nuclei of interest, with a common RF amplitude (nutation frequency) $\nu_1$ that matches a multiple of the geminal intra-pair $J$-coupling, i.e., $\nu_1 = n\, J^{intra}_{HH}$ with $n = 1$ for double- and $n = 2$ for single-

quantum SLIC. This leads to a population of the LLS through level anti-crossings (LACs). This population imbalance between states with different permutation symmetry is then allowed to relax during a delay $\tau_{rel}$. Since pairs of protons in $CH_2$ groups are chemically equivalent in achiral molecules (i.e., have the same chemical shifts), and, in the absence of couplings to heteronuclei, are often nearly magnetically equivalent, there is no need to suppress singlet-to-triplet leakage by transporting the sample into a region of low magnetic field, or by applying an RF field to sustain the imbalance. After a $T_{00}$ filter designed

to remove shorted-lived terms, a second SLIC pulse reconverts the remaining LLS back into observable magnetization for detection. In this work, single, double, and triple SLIC experiments were carried out to determine $T_{LLS}$, as shown by wavy arrows in Figure 2b and c.

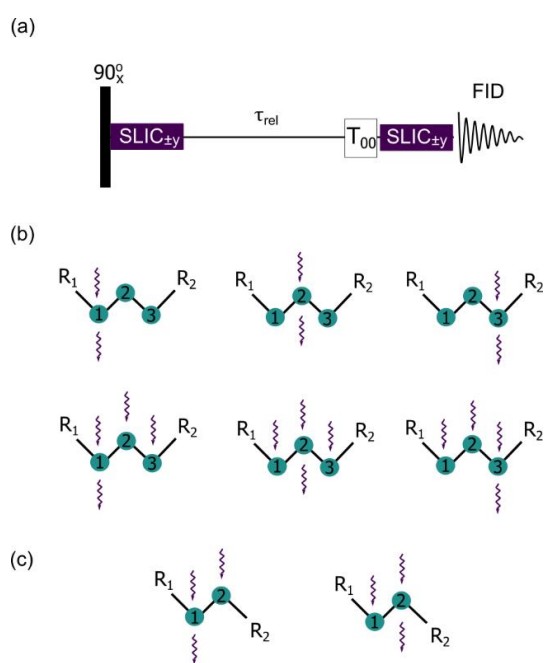

Figure 2. (a) Generic pulse sequence for single- and poly-spin-lock induced crossing (SLIC) where selective RF fields can be applied simultaneously to two or more $CH_2$ groups. (b) Six possible poly-SLIC experiments applied to molecules containing three $CH_2$ groups (I and II of Fig. 1a). The upper row shows three experiments with irradiation at a single offset for the creation of LLS and a single readout pulse applied to the offset of the first, second or third $CH_2$ group; the lower row shows three experiments using triple irradiation of all three $CH_2$ groups for LLS excitation, combined with a single readout SLIC applied to only one of the three $CH_2$ groups. (c) Two schemes with double SLIC excitation and single SLIC readout for compounds containing only two $CH_2$ groups (III and IV of Fig. 1a).

Titrations were performed by preparing a set of samples where all compounds except TEMPOL had fixed
concentrations. A stock solution with 40 mM of each compound was diluted by a factor 4 to obtain a final concentration of 10 mM for each compound. Each sample contained 10 mM of each compound in a 250 mM phosphate buffer at pH 7.0. A 20 mM TEMPOL stock solution was diluted in steps and added to yield final concentrations of 0.5, 1.0, 2.0, 3.0, 4.0, and 6.0 mM. The [1]H NMR spectra were recorded using a 500 MHz AVANCE Neo Bruker spectrometer with a 5mm iProbe at 298 K. Each sample contained a mixture of all four molecules, thus ensuring accurate comparisons of relaxation rates of different
molecules. The assigned [1]H NMR spectrum of the mixture is presented in Figure 1b. Typical signal decays due to LLS



relaxation as a function of the TEMPOL concentration are shown in Figure 3. Simulations of the contribution of different LLS terms to the observed signal were performed in SpinDynamica (Bengs and Levitt, 2018).

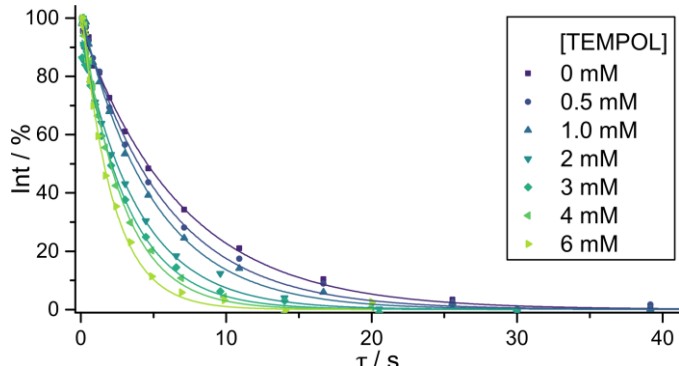

**Figure 3. Decay of LLS-derived signals of DSS (compound I) for different TEMPOL concentrations. LLS were excited and reconverted by irradiation with single SLIC pulses applied to $CH_2^{(1)}$ with an RF amplitude of 27 Hz to match the single-quantum level anti-crossing (SQ LAC). The solid lines correspond to mono-exponential fits.**

**Results and discussion**

**1.1 Comparison of relaxivities of long-lived states and of longitudinal magnetization: partly correlated random fields**

As apparent in Figure 4, both the longitudinal relaxation rate $R_1 = 1/T_1$ and the long-lived relaxation rate $R_{LLS} = 1/T_{LLS}$ depend linearly on the concentration of TEMPOL (in units of M or mol/L):

$$R_1 = R_1^{(0)} + r_1 \, [\text{TEMPOL}],$$
$$R_{LLS} = R_{LLS}^{(0)} + r_{LLS} \, [\text{TEMPOL}]. \tag{1}$$

The slopes $r_{LLS}$ and $r_1$ are known as *relaxivities* (in units of $M^{-1}s^{-1}$); the intercepts $R_1^{(0)}$ and $R_{LLS}^{(0)}$ are the rates determined in the absence of TEMPOL. Figure 4 shows that variations of $R_1$ between neighboring $CH_2$ groups within each molecule are much smaller than variation of those between different molecules. Whereas $T_1$ values of small molecules correlate with the molecular weight – the larger molecule, the shorter $T_1$ – this is not true for $T_{LLS}$. In the absence of TEMPOL, the longest $T_{LLS}$ of ca. 15 s was observed for compound III, whereas the shortest $T_{LLS}$ of ca. 5 s was found for compound IV, although their $T_1$ relaxation times and their molecular masses are roughly the same, so that their correlation times should be similar. The difference of $T_{LLS}$ may be explained by presence of 12 methyl protons in compound IV, which cause faster relaxation of LLS.

Wokaun and Ernst famously demonstrated that PRE is less efficient for relaxation of zero-quantum coherences than for single- and double-quantum coherences (Wokaun and Ernst, 1978). Tayler and Levitt demonstrated that a similar logic also applies to LLS: whereas longitudinal relaxation is enhanced by fluctuations of external local fields induced by unpaired electrons of radicals, an LLS involving two spins $\vec{I}_1$ and $\vec{I}_2$ is only relaxed by fluctuating external fields if these are *not* correlated. In general, the extent of correlation of the two fluctuating fields at the locations of the two spins $\vec{I}_1$ and $\vec{I}_2$ can be characterized




by the correlation coefficient $C = \langle \vec{B}_1 \cdot \vec{B}_2 \rangle / (B_1 B_2)$, where $B_i = \sqrt{\langle \vec{B}_i \cdot \vec{B}_i \rangle}$ is the mean (time-averaged) amplitude. Only the

*uncorrelated* part of the two fluctuating fields given by $\langle (\vec{B}_1 - \vec{B}_2) \rangle^2 = (B_1^2 + B_2^2 - 2\langle \vec{B}_1 \cdot \vec{B}_2 \rangle)$ contributes effectively to LLS

125  relaxation (Tayler and Levitt, 2011). The smaller the radical, the closer it can approach one of the two geminal protons, hence

the smaller the correlation coefficient $C$. It has been shown (Tayler and Levitt, 2011) that the ratio of relaxivities:

$$\kappa = r_{LLS}/r_1, \qquad\qquad (2)$$

is a characteristic measure of the correlation coefficient $C$; the smaller $\kappa$, the larger $C$. The experimental ratios $\kappa$ for the CH$_2$

group in the (chiral) dipeptide alanine-glycine varied in the range $0.5 < \kappa < 0.3$ depending on the size of the paramagnetic

agent (Tayler and Levitt, 2011). A similar ratio $\kappa = 0.36$ was observed for the CH$_2$ group in the terminal glycine residue of the

130  tripeptide Ala–Gly–Gly for PRE caused by triplet oxygen (Erriah and Elliott, 2019).

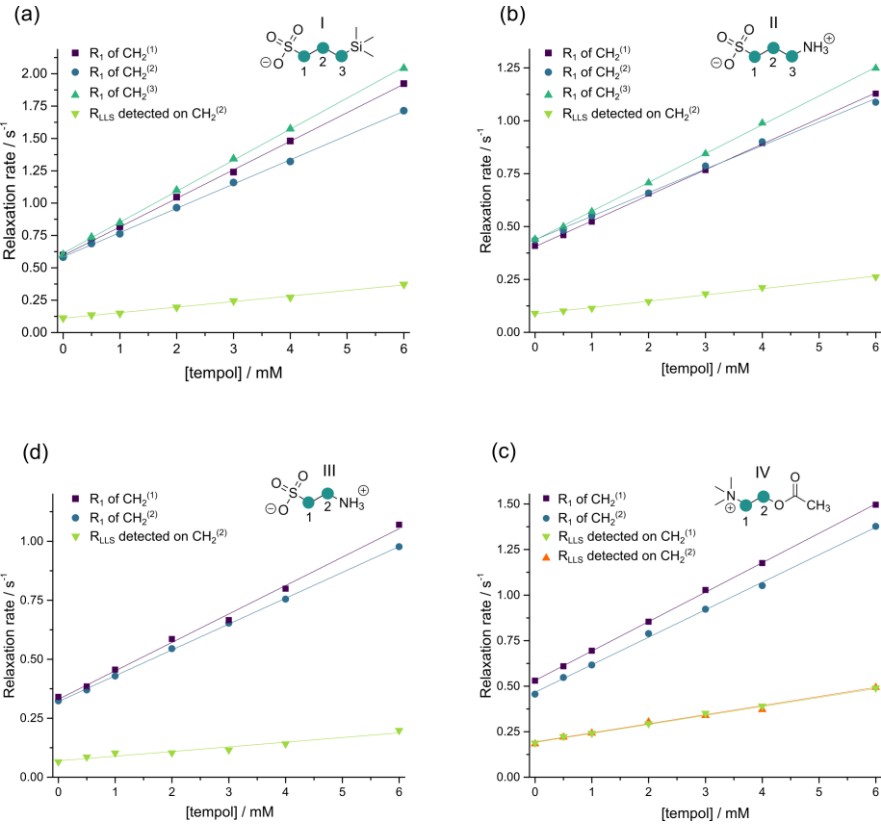

**Figure 4. Relaxation rates $R_1 = 1/T_1$ and $R_{LLS} = 1/T_{LLS}$ in CH$_2$ groups of four molecules I-IV as a function of the TEMPOL concentration. In (a) and (b), the LLS were excited by triple SLIC, in (c) and (d) by double SLIC, both with an RF amplitude of 13.5 Hz to match the double-quantum level anti-crossing (DQ LAC.) In all cases, the LLS were reconverted by single SLIC applied**

135  **to the CH$_2^{(2)}$ group, except for compound IV, where two sets of experiments with the reconversion performed at the CH$_2^{(1)}$ and at the CH$_2^{(2)}$ group. The relaxivities $r_1$ and $r_{LLS}$ correspond to the slopes of the linear regressions.**



In CH$_2$ chains excited by exploiting magnetic inequivalence in achiral molecules, the LLS can be delocalized over
several CH$_2$ groups. Relaxation of an LLS localized within an individual CH$_2$ group will contribute to the decay of a delocalized
LLS, so that one may expect the relaxivity of delocalized LLS to be more strongly affected by PRE than the relaxivity of a
(hypothetical) localized LLS. We must however remain cautious, all the more since the longitudinal magnetizations of
individual CH$_2$ groups may have a different relaxivities $r_1$. As we shall discuss below, the variations in the observed relaxivities
$r_{LLS}$ are not very large for different combinations of excitation and reconversion methods, and these intramolecular variations
are much smaller than the intermolecular differences between distinct compounds, so that one can estimate an average ratio of
relaxivities $\langle \kappa \rangle = \langle r_1 \rangle / \langle r_{LLS} \rangle$ for all CH$_2$ groups in a given molecule. Compounds I-IV feature average ratios $\langle \kappa_I \rangle \approx 0.22$,
$\langle \kappa_{II} \rangle \approx 0.23$, $\langle \kappa_{III} \rangle \approx 0.18$, and $\langle \kappa_{IV} \rangle \approx 0.32$ (see Table 1). Note the similarity of the ratios $\langle \kappa_{II} \rangle$ and $\langle \kappa_{III} \rangle$ obtained for
compounds that differ by only one CH$_2$ group. The LLS can be delocalized to a variable extent between all three CH$_2$ groups
in I and II, but always equally distributed between the two CH$_2$ groups in compounds III and IV.

**Table 1. Experimentally determined relaxation rates (s$^{-1}$) and relaxivities (M$^{-1}$s$^{-1}$). Standard errors determined from linear fits are shown in parentheses. For double SLIC, the RF amplitude was chosen to match double-quantum level anti-crossing (LAC) conditions, leading to different imbalances characterized by different decay rates $R_{LLS}^{(0)}$(SQ) with single SLIC excitation and single SLIC receonversion and $R_{LLS}^{(0)}$(DQ) with triple SLIC excitation and single SLIC reconversion.**

| Compound | $R_1^{(0)}$ | $R_{LLS}^{(0)}$(SQ) | $R_{LLS}^{(0)}$(DQ) | $r_1$ | $r_{LLS}$(SQ) | $r_{LLS}$(DQ) |
|---|---|---|---|---|---|---|
| I, CH$_2^{(1)}$ | 0.596(6) | 0.144(2) | 0.111(4) | 0.221(2) | 0.051(1) | 0.043(1) |
| I, CH$_2^{(2)}$ | 0.585(6) | 0.116(2) | 0.106(3) | 0.188(2) | 0.046(1) | 0.045(1) |
| I, CH$_2^{(3)}$ | 0.613(5) | 0.125(3) | 0.113(2) | 0.240(2) | 0.054(1) | 0.048(1) |
| II, CH$_2^{(1)}$ | 0.405(4) | 0.120(2) | 0.093(3) | 0.121(1) | 0.029(1) | 0.022(1) |
| II, CH$_2^{(2)}$ | 0.438(9) | 0.102(3) | 0.088(3) | 0.111(3) | 0.032(1) | 0.030(1) |
| II, CH$_2^{(3)}$ | 0.437(3) | 0.114(7) | 0.089(3) | 0.136(1) | 0.032(2) | 0.022(1) |
| III, CH$_2^{(1)}$ | 0.33(1) | - | - | 0.120(3) | - | - |
| III, CH$_2^{(2)}$ | 0.321(3) | - | 0.069(7) | 0.109(1) | - | 0.020(2) |
| IV, CH$_2^{(1)}$ | 0.532(4) | - | 0.194(3) | 0.162(1) | - | 0.050(1) |
| IV, CH$_2^{(2)}$ | 0.467(8) | - | 0.191(6) | 0.151(3) | - | 0.049(2) |

## 1.2 Implications for dissolution DNP

Even though delocalized LLS are less affected by TEMPOL than longitudinal magnetization, the observed decrease in $T_{LLS}$ is
undesirable in the context of d-DNP. Since the use of TEMPOL or other polarizing agents is mandatory for d-DNP
experiments, the question arises if it is worth scavenging TEMPOL after dissolution by addition of a reducing agent such as
sodium ascorbate (vitamin C) to extend $T_{LLS}$ after dissolution (Miéville et al., 2010, 2011). Note that the preparation of samples





comprising two types of beads is rather cumbersome and has not been attempted so far for bullet DNP. According to Miéville et al., the rate of the reduction of TEMPOL by sodium ascorbate may be slow on the time-scale of the transfer of the dissolved sample from the polarizer to the NMR magnet. Hence the reaction may not be entirely completed by the time the sample arrives in the spectrometer, and only a partial reduction of $R_{LLS}$ may be achieved. Scavenging by sodium ascorbate may be

accelerated ca. 100 times if one uses Frémy's salt instead of TEMPOL (Negroni et al., 2022). Several alternative approaches have been developed to remove radicals once DNP has been achieved. One approach is to use radicals obtained by UV irradiation of frozen pyruvic acid. These radicals are quenched as soon as the temperature increases (Eichhorn et al., 2013). One may also use radicals grafted into mesostructured silica materials (Gajan et al., 2014) or onto microporous polymers  (Ji et al., 2017; El Daraï et al., 2021). However, the small relaxivities presented in Table 1 suggest that scavenging may not be

necessary when using LLS for the transport and preservation of spin hyperpolarization.

### 1.3 Experiments and simulations for molecules with three CH₂ groups

It was shown (Sonnefeld et al., 2022b) that for the excitation of LLS in systems with $n = 3$ neighboring $CH_2$ groups, i.e., with $2n = 6$ spins, there are 7 orthogonal LLS product operators that can be created, with 7 coefficients $\lambda_i$ that depend on the excitation scheme:

$$\hat{\sigma}_{LLS} = \left(-\lambda_{AA'}\hat{I}^A \cdot \hat{I}^{A'} - \lambda_{MM'}\hat{I}^M \cdot \hat{I}^{M'} - \lambda_{XX'}\hat{I}^X \cdot \hat{I}^{X'}\right)$$
$$\left[-\lambda_{AA'MM'}\left(\hat{I}^A \cdot \hat{I}^{A'}\right)\left(\hat{I}^M \cdot \hat{I}^{M'}\right) - \lambda_{AA'XX'}\left(\hat{I}^A \cdot \hat{I}^{A'}\right)\left(\hat{I}^X \cdot \hat{I}^{X'}\right) - \lambda_{MM'XX'}\left(\hat{I}^M \cdot \hat{I}^{M'}\right)\left(\hat{I}^X \cdot \hat{I}^{X'}\right)\right] \quad (3)$$
$$-\lambda_{AA'MM'XX'}\left(\hat{I}^A \cdot \hat{I}^{A'}\right)\left(\hat{I}^M \cdot \hat{I}^{M'}\right)\left(\hat{I}^X \cdot \hat{I}^{X'}\right),$$

Here A and A' denote the two protons of the $CH_2^{(1)}$ group, M and M' those of the middle $CH_2^{(2)}$ group, and X and X' those of the terminal $CH_2^{(3)}$ group. This equation gives a general form of the density operator obtained after poly-SLIC, containing all long-lived terms found by numerical solution of the Liouville-von-Neumann equation. In addition to three bilinear terms, one encounters four higher terms that contain products of 4 and 6 spin operators. In principle, each term in Eq. (3) can decay with a different rate, so that one could distinguish up to 7 distinct rates $R_{LLS}^{(\mu)}$ with μ = AA', MM', XX', AA'MM', AA'XX',

MM'XX' and AA'MM'XX'. As was mentioned above, each term can have a different amplitude and can contribute with a different weight to the observed signal.

In systems such as compounds III and IV with only two $CH_2$ groups only one LLS can be excited:

$$\hat{\sigma}_{LLS} = \left(-\lambda_{AA'}\hat{I}^A \cdot \hat{I}^{A'} - \lambda_{XX'}\hat{I}^X \cdot \hat{I}^{X'}\right) - \lambda_{AA'XX'}\left(\hat{I}^A \cdot \hat{I}^{A'}\right)\left(\hat{I}^X \cdot \hat{I}^{X'}\right), \quad (4)$$

The coefficients of the first two bilinear terms are always equal, i.e., $\lambda_{AA'} = \lambda_{XX'}$ while the 4-spin term is always proportional to these bilinear terms, with a weight $\lambda_{AA'XX'} = 8/3\ \lambda_{AA'}$ (Sonnefeld et al., 2022a). This state corresponds to the imbalance

between the singlet-singlet state and the triplet-triplet manifold and is therefore expected to decay monoexponentially. In two sets of complementary experiments performed for compound IV, the experimental relaxation rates were indeed found to be indistinguishable, as can be seen by comparing the red triangles and the green inverted triangles in Figure 4d.



In compounds I and II however, which contain three adjacent $CH_2$ groups, different SLIC excitation schemes lead to population of different LLS, with different coefficients $\lambda_{LLS}^{(\mu)}$ in Eq. (3). There are 9 different ways of exciting miscellaneous

LLS and 9 different ways of reconverting them, giving 81 possible experimental combinations. In order to investigate the relaxivities of these different LLS which may have different decay rates $R_{LLS}^{(\mu)}$ and relaxivities $r_{LLS}^{(\mu)}$, we performed 6 different poly-SLIC experiments with different SLIC pulses for excitation and reconversion, and indeed found different LLS lifetimes (Figure 5). Depending on the excitation and reconversion scheme used, there are pronounced differences between the relaxivities $r_{LLS}$ within one and the same molecule.

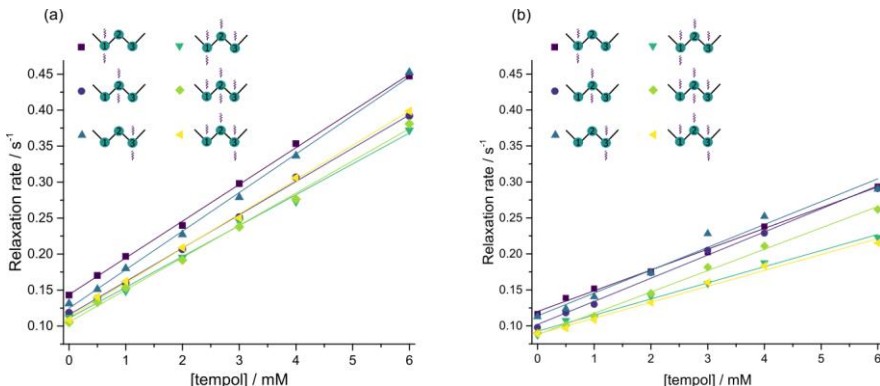

**Figure 5. Decay rates $R_{LLS} = 1/T_{LLS}$ of long-lived states in $CH_2$ groups in (a) DSS (I) and (b) homotaurine (II), each containing three $CH_2$ groups, as a function of the TEMPOL concentration. Six different poly-SLIC experiments with distinct excitation and reconversion methods were performed for each molecule, as indicated by wavy arrows. The relaxivities $r_{LLS}$ correspond to the slopes of the linear regressions.**

We calculated the contributions of each of the 7 terms to the observable LLS-derived signals, after two consecutive transformations $\hat{I}_z^{in} \rightarrow \hat{\sigma}_{LLS} \rightarrow \hat{I}_x^{obs}$ (see Figure 6). For each excitation scheme used in this work, all 7 coefficients $\lambda_\mu^{M \rightarrow LLS}$ corresponding to the 7 terms in Eq. (3), as well as all 7 reconversion coefficients $\tilde{\lambda}_\mu^{LLS \rightarrow M}$. The coefficients are calculated according to:

$$\lambda_\mu^{M \rightarrow LLS}(\hat{I}_z^{in} \rightarrow \hat{\sigma}_{LLS}) = \frac{\text{Tr}\{\hat{P}_\mu^\dagger \cdot \hat{\sigma}_{LLS}\}}{\text{Tr}\{\hat{P}_\mu^\dagger \cdot \hat{P}_\mu\}},$$

$$\tilde{\lambda}_\mu^{LLS \rightarrow M}(\hat{P}_\mu \rightarrow \hat{I}_{x,\mu}^{obs}) = \frac{\text{Tr}\{\hat{I}_x^\dagger \cdot \hat{I}_{x,\mu}^{obs}\}}{\text{Tr}\{\hat{I}_x^\dagger \cdot \hat{I}_x\}},$$

(5)

Here index μ corresponds to one of the 7 LLS terms in Eq. (3), the operator $\hat{P}_\mu$ represents LLS μ-th term, $\hat{I}_z^{in}$ is the initial

magnetization of the excited spins, $\hat{I}_x$ is the transverse magnetization of the observed spins after reconversion, and $\hat{I}_{x,\mu}^{obs}$ is the transverse magnetization obtained after reconversion of only the μ-th term $\hat{P}_\mu$ instead of the full $\hat{\sigma}_{LLS}$. The observed signal $S_\mu$ stemming from the μ$^{th}$ term is determined by the product of two coefficients $\lambda_\mu^{M \rightarrow LLS}$ and $\tilde{\lambda}_\mu^{LLS \rightarrow M}$ for a given combination of excitation and reconversion SLIC pulses. These contributions are shown in Figure 6. The sum of all 7 amplitudes of all combinations of excitation and reconversion SLIC schemes was normalised to one. These graphs show how the LLSs are





delocalized across the spin system comprising $n = 3$ neighboring CH$_2$ groups. We only consider coherent spin dynamics during excitation and reconversion, neglecting possible redistributions of LLS due to Overhauser-type cross-relaxation effects, and neglecting zero quantum coherences.

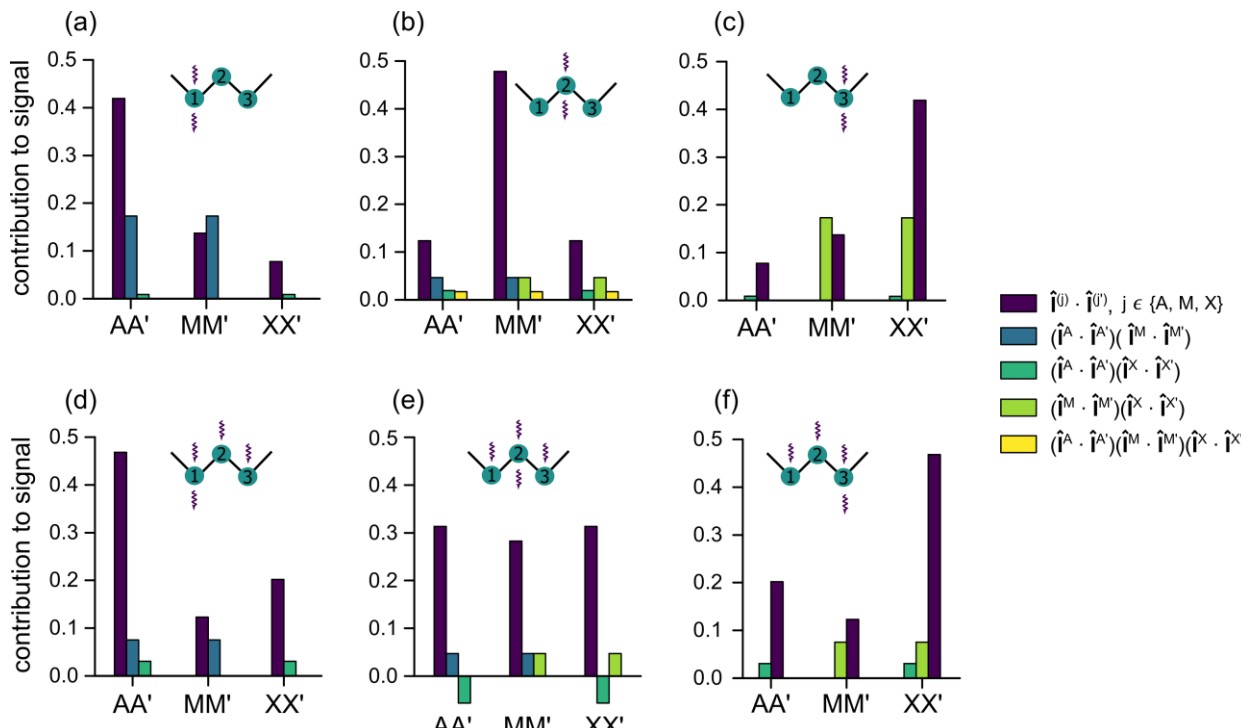

**Figure 6.** Calculated contributions of the seven different LLS terms $\hat{P}_\mu$ in the density operator of Eq. (3)) to the observed signals for
all 6 different single- and poly-SLIC experiments used in this work to determine the relaxivities $r_{LLS}^{(a)}$ in DSS (I) and homotaurine (II). The histograms show the products $\lambda_\mu^{M \to LLS} \lambda_\mu^{LLS \to M}$ of the coefficients of LLS excitation and reconversion methods. The normalisation ensures that the sum of all products of coefficients is equal to 1. Experiments with triple SLIC excitation and single SLIC reconversion applied to the middle CH$_2$ group (e) provide LLS states that are almost evenly distributed among all three CH$_2$ groups, whereas the other experiments provide access to LLS states that are in part localised on the group where the reconversion
SLIC pulse is applied. The (yellow) six-spin term is negligible except for case (b).

Note that a *single* SLIC pulse applied at the chemical shift of *any* of the three CH$_2$ groups results in the excitation of a delocalized state, which is predominantly (but not exclusively) associated with the irradiated pair. By using triple SLIC excitation and single SLIC reconversion applied to the middle CH$_2$ group, one can excite a fairly even distribution of the LLS involving all $2n = 6$ coupled spins. For compound II, the most strongly delocalized state features the largest relaxivity $r_{LLS}$.
For compound I, however, the largest relaxivities were obtained for experiments where the largest contribution to the observed signal came from the terminal group CH$_2$$^{(3)}$, which is closest to the trimethylsilane group. This group has also the largest longitudinal relaxivity $r_1$, as can be seen in Figure 4a. Detailed calculations of the relaxation superoperator might help to rationalize the experimental results obtained here.

**MAGNETIC RESONANCE**
Discussions

**Conclusions**

The relaxation rates of various long-lived states and of the longitudinal magnetization of DSS, homotaurine, taurine and acetylcholine were measured as a function of the concentration of the radical TEMPOL. In all cases, the relaxivities $r_{LLS}$ are lower by about a factor 3 compared to the relaxivities $r_1$. This implies that the effect of paramagnetic relaxation enhancement on LLS due to TEMPOL during d-DNP sample transfer might be limited. Furthermore, the LLS relaxivity was studied depending on different SLIC excitation and reconversion schemes. The results support simulations that show that different

LLS are excited depending on the number of adjacent methylene units in the molecule and the SLIC sequence used. SLIC methods have also been shown to be efficient for other achiral molecules containing neighboring $CH_2$ groups, such as dopamine, taurine and γ-aminobutyric acid (GABA), ethanolamine, and β-alanine (Sonnefeld, 2022a). All of these molecules contain aliphatic chains, so that the effects of paramagnetic polarizing agents like TEMPOL should be similar to what is reported in this work.

**Data availability**

All original NMR data obtained for this paper is available through the Zenodo repository under https://doi.org/10.5281/zenodo.7432635

**Conflict of interest**

Geoffrey Bodenhausen is a member of the editorial board of Magnetic Resonance Ampere. The peer-review process was

guided by an independent editor, and the authors have also no other competing interests to declare.

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
