# Peer review of "Paramagnetic relaxivity of delocalized long-lived states of protons in chains of CH2 groups"

_Magnetic Resonance, 2022_

## Author Comment (AC1)

Département de Chimie Ecole Normale Supérieure 24 Rue Lhomond, Paris 75005, France Dr Kirill Sheberstov

To the handling Editor "Magnetic Resonance"

Paris, January 18th 2023

Re: Revision mr-2022-23

"Paramagnetic relaxivity of delocalized long-lived states of protons in chains of CH2 groups" by Aiky Razanahoera, Anna Sonnefeld, Geoffrey Bodenhausen, Kirill Sheberstov

We thank the reviewers for their valuable comments.

We have thoroughly revised our paper in the light of the comments of the reviewers, which *are reproduced in italics below*, while our responses are in blue Roman script:

**Reviewer: RC1 (Malcolm Levitt)**

This is a very nice report continuing the interesting research line of long-lived states in chains of CH2 groups, initiated recently by this group. The authors explore the relaxivities of LLS in a variety of CH2 groups, and also different LLS groups within the same molecule, with respect to the common DNP agent TEMPOL. The paper is well-reasoned and well-presented and is highly suitable for publication. I have a few minor comments:

The introduction on page 1 is slightly misworded, in my view. The singlet-triplet population imbalance for spin-1/2 pairs, also known as singlet order, is just one special case of a long-lived state. Singlet order is indeed immune to intramolecular dipole-dipole relaxation, but that is not necessarily the case for more complex LLS, including some of the ones discussed later on in the paper.

We have clarified this as follows "In coupled pairs of spins with  $I = \frac{1}{2}$ , such long-lived states (LLS) correspond to population imbalances between singlet and triplet states (Carravetta and Levitt, 2004; Carravetta et al., 2004) that are immune to intra-pair dipole-dipole interactions, which for pairs of protons are normally the dominant cause of longitudinal relaxation. In larger systems, LLS may involve four, six or more spins, all these states are weakly affected by dipolar relaxation (Hogben et al., 2011)."

on page 2 it is stated that fast proton relaxation is the reason for d-DNP being performed mainly on 13C and 15N nuclei: This is true in part, but an even more important reason, in some contexts, is the interference from strong proton background signals.

We have clarified "proton relaxation is fast so that the level of hyperpolarization suffers during dissolution and transfer, which is one of the reasons why d-DNP is more often used for 13C or 15N rather than for protons" (...) "After converting proton LLS back into proton magnetization, only proton signals of interest are observed, while the background is suppressed."

Presumably the buffer solution is aqueous. This should be stated, and if possible, the precise composition of the buffer should be given.

We have clarified "A stock solution of phosphate buffer (70 mM  $KH_2PO_4$  and 130 mM  $K_2HPO_4$ ) was prepared in  $D_2O$  and diluted by factor 4 in each sample. A 20 mM TEMPOL stock solution was diluted in steps and added to yield final concentrations of 0.5, 1.0, 2.0, 3.0, 4.0, and 6.0 mM."

In Fig.3 the signals are presumably normalized such that "100%" is the first point in the decay. This should be stated.

We have clarified in the caption to Fig. 3: "The solid lines correspond to mono-exponential fits, **scaled to begin at 100%**."

The statement that PRE is less effective for ZQCs than other coherences is usually true, but at least in principle, anti-correlation of the fluctuating fields could cause the opposite relationship.

The correlation function *C* that we defined in Eq. (2), inspired by Tayler and Levitt (2011), takes into account the possibility of anti-correlated fluctuations, which would lead to a negative sign of the correlation function and cause faster relaxation of the LLS and supposedly of ZQC. There seems to be no reason to amend this definition.

In common with numerous others, the paper confuses "rates" (which become smaller as a process approaches equilibrium) with "rate constants" which (as the name implies) are time-independent for an exponential decay. So most of the reported "rates" in the paper are actually "rate constants".

We have replaced "rates" by "rate constants" in several occurrences.

The discussion on the long-lived states in chains of three CH2 groups is particularly interesting. I much appreciated Figure 6.

As the authors state, a random-field model may be used to relate the rate constant of singlet order relaxation to the correlation factor for fluctuating random fields at the two nuclear sites. I wonder what the corresponding expressions would be for the "delocalised" long-lived states explored by the authors. Many pairs of correlated fields could be involved. Are such rate constants dominated by the "least correlated pair" of fluctuating random fields?

A stimulating question that we cannot address in this work but that we will keep in mind.

**Reviewer: 2 (Alexej Jerschow)**

This is very interesting work, presenting the surprising results that delocalized long lived states can be longer lived than T1 by factors of 3-5 in systems with paramagnetic interactions. The context is DNP, where radicals would typically be present, and hence these findings can be of practical use as well. The insights gained with respect to the sizes of the paramagnetic agents and the simulations of Fig. 6 are particularly useful.

Here are my minor comments:

(1) The text refers to single, double, and triple SLIC sequences. Perhaps this requires a minor clarification: do you mean single, double, and triple frequency irradiation? One could also read this as several cycles of SLIC pulses.

On the first occurrence of this expression, on line 104, we have specified: In this work, **SLIC experiments** with single, double, and triple irradiation (henceforth called "single, double, and triple SLIC experiments" for simplicity) were carried out to determine  $T_{LLS}$ , as shown by wavy arrows in Error! R eference source not found.b and c.

(2) It would seem suitable to cite our recent work on paramagnetic relaxation (and other mechanisms) of singlet order by experiment and MD simulations: Kharkov et al, Phys. Chem. Chem. Phys., 2022, 24, 7531-7538, <a href="https://pubs.rsc.org/en/content/articlehtml/2022/cp/d1cp05537b?casa\_token=gMiUwO-MKY0AAAAA:N3Uh0TjOIoCxkD6BfVIFo1NoabXKCWK6mVqq5XN\_G5F\_ntohz0qsuLv0LAxW4veoIRTxIBzBN0DLATE">https://pubs.rsc.org/en/content/articlehtml/2022/cp/d1cp05537b?casa\_token=gMiUwO-MKY0AAAAA:N3Uh0TjOIoCxkD6BfVIFo1NoabXKCWK6mVqq5XN\_G5F\_ntohz0qsuLv0LAxW4veoIRTxIBzBN0DLATE</a>

We have inserted a citation of this paper on line 52: "On the other hand, the relaxation of LLS can be due to mechanisms such as dipolar couplings to solvent nuclei, even with low gyromagnetic ratios, and to paramagnetic species (Kharkov et al., 2022). "

**Reviewer: 3 (Anonymous Referee #3)**

Razanahoera et al. present the results of experimental investigations into long-lived nuclear spin states (LLSs) excited in delocalized pairs of protons, i.e.,  $CH_2$  groups, of several molecules and within the chain of the same molecule. The context of their exploration was relaxivity from the common 1H d-DNP agent TEMPO dissolved in solution at typical post-dissolution radical concentrations. The manuscript is well-written, well-thought out and contains clear figures. The work is interesting and suitable for publication after minor corrections.

**Major comments:**

1/ The authors frequently refer to relaxation rates and relaxation times. However, this is not what is gleamed from the experimental results. These are relaxation rate constants and relaxation time constants

. This is especially true in this case since mono-exponential decay curves are observed. This comment was also pointed out by Reviewer 1 and should be corrected throughout the manuscript.

We have replaced "rates" by "rate constants" in several occurrences.

2/ In the abstract, the authors note that "...the yield of conversion of observable magnetization into LLS and back are on the order of 10% or less...". This is likely quite an unfair lower bound to use as motivation. Typically, 30-45% of the 2/3 theoretical maximum can be achieved with such a rf-pulse sequence. Furthermore, the contribution to signal is given, whilst the conversion yield vs. observable magnetization is not. A typical value should be quoted in the manuscript.

The theoretical maximum efficiency in a four-spin system in an aliphatic chain was previously calculated to be 14% for single SLIC irradiation and 28% for double SLIC irradiation [Sonnefeld et. al., Sci. Adv., 8, eade2113]. Our estimate « on the order of 10% or less » is based on empirical observations, which as expected tend to be below theoretical predictions. We have inserted the word "empirical" in the abstract, and clarified that this factor refers to the combination of excitation and reconversion: "Since the empirical yield of the conversion and reconversion of observable magnetization into LLS and back is on the order of 10% "... We also added two sentences to clarify this in lines 110-115.

3/ There are a number of important works from the LLS community that are not cited. Ratio of  $T_{LLS}/T_1$ : Angew. Chem. 2015, **127**, 3811-3814. J. Am. Chem. Soc. 2012, **134**, 17494-17497.  $T_{00}$  Filter: J. Am. Chem. Soc. 2013, **135**, 2120-2123. These should be incorporated into the manuscript.

We have inserted these references on lines 60-65. We have also inserted a reference to our own early work on proton LLS, since for many years our group actually held the world record in terms of  $T_{LLS}/T_1$  ratios, in a pair of exocyclic diastereotopic protons in a partly deuterated saccharide, where  $T_{LLS}/T_1 = 36$ . (See Singlet-State Exchange NMR Spectroscopy for the Study of Very Slow Dynamic Processes. R. Sarkar, P. R. Vasos, and G. Bodenhausen, J. Am. Chem. Soc. 129, 328-334 (2007).)

**Minor comments:**

1/ The authors mention that LLSs in homonuclear spin-1/2 pairs are immune to intra-pair dipole-dipole interactions. It is more likely to be correct to say intra-pair dipole-dipole relaxation.

In our humble opinion, the intra-pair dipole-dipole interactions are responsible for the intra-pair dipole-dipole relaxation.

2/ Were the samples degassed? Would the results differ significantly is the samples were degassed? The samples were not degassed, as we have now specified on line 113: "...without removing paramagnetic oxygen by degassing". Degassing would probably lead to a minor extension of the TLLS, but this is hardly relevant for applications to dissolution DNP.

3/ In Table 1, the ratios of the relaxation rate/time constants would also be nice to see.

Some of the ratios are specified in the text and straightforward to calculate from the table 1. Filling in this information in the table would require addition of 4 additional columns which would make the table unwieldly so we rather not include those ratios.

4/ It would be nice to know the number of scans used for experiments.

We have inserted in the material and method section: "were obtained by adding 16 signals (for experiments with single SLIC irradiation) and 8 signals (for experiments with multiple SLIC irradiation) at 500 MHz".

5/ "reconversion" is incorrectly spelt in the caption of Table 1.

This typo has been corrected.

6/ "relaxation" is incorrectly spelt in the short summary.

We thank the reviewer for noticing this typo, we will correct it upon resubmission.

We thank reviewers for their suggestions.

Yours sincerely,

Dr Kirill Sheberstov Ecole Normale Supérieure 24 rue Lhomond, 75231 Paris cedex 05, France